# Loop Extrusion Machinery Impairments in Models and Disease

**DOI:** 10.3390/cells13221896

**Published:** 2024-11-17

**Authors:** Anastasiya Ryzhkova, Ekaterina Maltseva, Nariman Battulin, Evelyn Kabirova

**Affiliations:** 1Institute of Cytology and Genetics, 630090 Novosibirsk, Russia; anastryzhk@gmail.com (A.R.); battulin@gmail.com (N.B.); 2Department of Genetics and Genetic Technologies, Sirius University of Science and Technology, 354340 Sirius, Russia; emaltseva36@gmail.com; 3Department of Natural Sciences, Novosibirsk State University, 630090 Novosibirsk, Russia

**Keywords:** cell biology, chromatin structure, 3D genome, SMC, cohesin, condensin

## Abstract

Structural maintenance of chromosomes (SMC) complexes play a crucial role in organizing the three-dimensional structure of chromatin, facilitating key processes such as gene regulation, DNA repair, and chromosome segregation. This review explores the molecular mechanisms and biological significance of SMC-mediated loop extrusion complexes, including cohesin, condensins, and SMC5/6, focusing on their structure, their dynamic function during the cell cycle, and their impact on chromatin architecture. We discuss the implications of impairments in loop extrusion machinery as observed in experimental models and human diseases. Mutations affecting these complexes are linked to various developmental disorders and cancer, highlighting their importance in genome stability and transcriptional regulation. Advances in model systems and genomic techniques have provided deeper insights into the pathological roles of SMC complex dysfunction, offering potential therapeutic avenues for associated diseases.

## 1. Experimental Models of the Loop Extrusion Machinery

Proteins of the loop extrusion machinery (see Figure 1 and the next chapter for a description of the protein composition of the complexes) are known to shape mitotic chromosomes, as initially, their role in the separation of sister chromatids was discovered [1,2]. But a mechanism for how they provide chromosome structure not only during cell division, but also during interphase, as well as their role in the processes related to chromatin structure, such as transcription and gene regulation, remained implicit due to the challenges of creating an experimental model.

SMCs are essential for early embryonic development. Consequently, homozygous deletions of the genes encoding SMCs cause early embryonic lethality in mice, while heterozygous deletions cause variable abnormal phenotypes. *SMC3* heterozygous mice showed reduced body weight and craniofacial defects [3]. Craniofacial defects were also detected in *Nipbl* and *Mau2* mutant mice—models of Cornelia de Lange Syndrome [4,5]. *Rad21* heterozygous mice were shown to have higher sensitivity to whole-body irradiation, suggesting DNA repair defects [6].

Impairment of SMCs could affect mouse fertility. Mice lacking subunits of meiotic cohesin are sterile [7,8,9,10]. Condensin II dysfunction has been associated with hybrid incompatibility in female meiosis in mice [11]. Due to divergence in condensin II abundance on chromosomes between different species, an amount of condensin II in hybrid oocytes could be insufficient, causing chromosome decondensation, centromere stretching, and mis-segregation. The NCAPG2 subunit of condensin II happens to act as a limiting factor to form a functional condensin II complex in hybrids, with its overexpression rescuing decondensation and egg aneuploidy. Notably, condensin II is reduced in *Mus musculus domesticus* x *Mus spretus* hybrid oocytes, but not in their somatic cells, enabling normal development, but also leading to fertility issues.

Overall, mouse models have revealed that SMC complexes are extensively utilized for different biological processes. But the details of their role in these processes are difficult to distinguish. Establishment of the 3C-based methods (chromosome conformation capture) provided a unique tool enabling studies of chromatin spatial organization. Together with the discovery of topologically associating domains (TADs) in 2012 [12,13], this provoked a burst of interest towards the role of loop extrusion machinery in shaping chromosomes. Thus, new approaches to overcome the necessity of SMC complexes in cell division and, consequently, the lethality of homozygous deletions were implemented in experimental models.

One approach is the conditional inactivation of a gene encoding a subunit of an SMC complex in non-dividing cells, such as hepatocytes [14,15]. It requires an integration of two *loxP* sites flanking a target gene and a tissue-specific Cre recombinase driver in mice. Then, tamoxifen injection in mice induces Cre recombinase activation in the specific cell type, leading to deletion of the target gene. Implementing the technique helped us to discover that deletion of the cohesin-loading factor *Nipbl* in mouse liver leads to displacement of cohesin from chromatin [14]. Consequently, it caused a global reorganization of 3D chromatin organization, with TADs vanishing. It was also utilized to reveal that chromatin is organized into TADs and loops in mouse zygotes through the mechanism of cohesin-mediated loop extrusion [16]. Thus, conditional inactivation of SMCs has provided a great advantage in in vivo studies, though with limitations: the protein degradation is slow, as it takes about five days before primary cell cultures with gene inactivation can be obtained from animals, and potentially asynchronous, as genetic deletion might not occur simultaneously in all cells.

Notably, a knockout of cohesin loader *WAPL* could be obtained in a cell culture. Loss of *WAPL* has been shown to cause p53-dependent cell cycle arrest [17,18]. This feature inspired the usage of a p53-deficient cell culture to restore normal cell proliferation in *ΔWAPL*. For instance, WAPL depletion was successful in mESCs with decreased activity of the p53 pathway [19] and cells co-depleted of p53 [18,20].

Other approaches involve blocking transcription of a target gene without genome modifications by CRISPR interference and preventing mRNA translation of a target gene by RNA interference. Although these methods may result in asynchronous or incomplete depletion, they are easy to use and still suitable to tackle a variety of problems.

But a prevalent approach to study the function of the loop extrusion machinery is inducible degron technologies. They present unique advantages of homogenous, rapid (within 30–60 min), and reversible (with an inducer removal) protein degradation. A degron system employs an E3 ligase to induce polyubiquitination of a target protein, which triggers its degradation by the proteasome. Distinct degradation strategies utilize different E3 ligases and small molecules. Auxin-inducible degron (AID) is a three-part system that includes exogenous E3 ligase (TIR1 from plant *Oryza sativa*), target protein genetically tagged with AID, and an inducer, auxin [21]. Upon auxin’s introduction to the cells, genetically modified for expression of TIR1 and target protein-AID (degron), auxin binds to TIR1, promoting polyubiquitination of the dergron. Though efficient, the system has a drawback of leaky degradation, that is, without auxin induction. The authors overcame the issue by implementing a bump-and-hole approach, which employs an OsTIR1(F74G) mutant and a 5-phenyl-auxin, thus creating an AID version 2 system without leaky degradation [22]. FKBP12/dTAG is a two-part system requiring an inducer (dTAG) that can bind an endogenous E3 ligase and a degron tag (FKBP12^F36V^) introduced to the target protein [23]. Proteolysis-targeting chimera (PROTAC) is a one-part system that employs inducers engineered to directly link an endogenous E3 ligase to the target protein, avoiding the need for any genetic engineering [24]. As for SMCs studies, AID version 2 system is the most widely used.

It is worth mentioning that the studies in the field of 3D genomics often combine both experimental and computational models [25]. For example, in order to reveal loop extrusion, mechanism polymer simulations and computational predictions were utilized to expand the experimental data [26].

## 2. Structure and Function of the Loop Extrusion Machinery

A functional form of SMCs is in a complex with non-SMC proteins. The composition of these complexes are evolutionarily conserved from bacteria to humans, for an evolution overview, see [27]. Eukaryotes have four main loop extrusion complexes (Figure 1a): cohesin, which mediates interphase chromosome organization and sister chromatid cohesion; two types of condensin, which form X-shaped mitotic chromosomes; and SMC5/6, which participates in DNA damage repair and replication. All of these complexes share a similar tripartite ring structure formed by a dimer of SMC proteins and a kleisin protein, to which additional subunits attach.

Extrusion complexes may be loaded on chromatin directly (condensin I, SMC 5/6) or mediated by other proteins (cohesin, condensin II). A balance of cohesin is maintained by an interplay of specific loading and unloading factors binding to the kleisin subunit (Figure 1b), for a mechanism, see review [28]. Nipped-B-like protein (NIPBL) stabilized by MAU2 is responsible for cohesin loading on chromatin and activates cohesin translocation along DNA. WAPL bound to PDS5 counteracts NIPBL/MAU2 and mediates cohesin’s release from chromatin.

Cohesin acetylation additionally regulates chromatin loop extension. The acetyltransferase ESCO1 acetylates the cohesin SMC3, which in turn facilitates PDS5A binding to cohesin, and as a result limits the length of chromatin loops, while the deacetylase HDAC8 promotes loop enlargement [29]. Cohesin acetylation appears to control loop extension independently of its canonical role in protecting cohesin from WAPL during the S phase of the cell cycle.

CCCTC-binding factor (CTCF) blocks diffusive cohesin, acting as an anchor. On the one side, CTCF employs its zinc finger domain to bind to a specific site in the genome; on the other side, CTCF binds to cohesin subunits STAG1 or STAG2 [30]. Convergent CTCF-binding sites stabilize cohesin-mediated loop extrusion—the resulting structure is called a topologically associating domain (TAD). Since cohesin-mediated loop extrusion is a dynamic process, the structure of a TAD varies in single cells [31,32,33]. In some cells, cohesin extrudes a larger portion of chromatin than in others by the moment of cell fixation required to detect a 3D genome. But the average 3D genome across many cells of the same cell type is conservative, with minor changes across different cell types according to their specific functions [34,35]. Moreover, certain 3D patterns, such as the second diagonal on Hi-C maps of erythroid cells, are conservative across vertebrates [36]. TADs are believed to contribute to gene regulation by bringing an enhancer into proximity with their target promoters. Therefore, disruptions to TAD integrity could cause dramatic regulation rewiring, with enhancer hijacking initiating abnormal gene expression. This function has been proven by a number of studies disrupting the integrity of TADs [37,38,39,40,41,42] or discovering consequences of the TAD disruptions occurring in patients [43,44,45] or evolution [46]. However, the efficiency of TADs’ impact on gene regulation is restricted by the local epigenetic landscape [35,47].

Condensins were once believed not to require additional factors for loading, but recently, a protein, M18BP1, was identified which mediates condensin II loading on chromatin [48]. Furthermore, there is a mechanism to ensure appropriate timing for condensin loading. Kleisin subunits are responsible for restricting precocious binding of condensins on chromatin, either directly through its N-terminal tail in the case of condensin I [49] or by binding to MCPH1, which counteracts M18BP1, in the case of condensin II [50]. Deletions of either kleisin’s N-terminal tail or the *MCPH1* gene lead to the acceleration of condensin loading and mitotic chromosome assembly.

The SMC5/6 complex utilizes its multiple subunits for chromatin binding and demonstrates ssDNA-binding ability [51].

## 3. Dynamics of the Loop Extrusion Machinery in Cell Cycle

The three-dimensional structure of chromatin is highly dynamic and undergoes a number of changes during the cell cycle. During the interphase, chromatin consists of compartments and TADs, which are formed by chromatin loops. TADs participate in regulating gene expression by controlling spatial interactions with enhancers and other regulatory elements [52]. During mitosis, the chromatin undergoes extensive compaction and loses its transcriptional activity. This is achieved through the formation of numerous condensed chromatin loops. Different components of the loop extrusion machinery are responsible for all these changes. Therefore, their localization changes during the cell cycle according to the function they perform (Figure 2). A study on *Xenopus* eggs [53] showed that depletion of cohesin, but not of condensins I and II, leads to a dramatic decrease in loop extrusion in the interphase. Conversely, the depletion of condensins I and II, but not cohesin, leads to a decrease in loop extrusion in the metaphase. In addition, the mechanism of loop extrusion differs during the cell cycle. Cohesin extrudes loops symmetrically in the interphase, while condensins I and II do this asymmetrically during the metaphase.

Firstly, let us look at the cohesin dynamics in the cell cycle. Cohesin is present in the nucleus throughout the interphase, and some of it remains in mitosis up to the anaphase, where it participates in the proper segregation of sister chromatids.

During the transition from cell division to the interphase, an intermediate chromatin state is formed, which is almost completely devoid of loops and components of the loop extrusion machinery. This state occurs during the telophase [54,55]. After that, during cytokinesis, cohesin is actively loaded onto chromatin, and the boundaries of TADs and compartments begin to recover even before the cell enters the G1 phase. Cohesin exists in two isoforms depending on whether it contains a STAG1 (SA1) or a STAG2 (SA2) subunit. Cohesin-SA2 represents the majority of the total cohesin pool (75%), while cohesin-SA1 represents a smaller portion (25%) [56]. Despite the fact that the two SA subunits have more than 70% sequence identity in the central part of the protein [57], recent studies have shown that cohesin-SA1 and cohesin-SA2 perform different functions in a cell. Thus, a model has been proposed that suggests that cohesin isoforms form interphase loops according to a mechanism of nested loops [58].

After cell division is complete, cohesin-SA1 is rapidly (within 10 min) imported into the chromatin intermediate, which at the same time also rapidly binds to CTCF proteins [55,58]. Imported cohesin-SA1 forms the first CTCF-limited long loops of TADs necessary for the initiation of housekeeping gene expression [58,59]. These loops are long-lived, since cohesin-SA1 is more stable and protected from the WAPL unloader due to the ESCO1-mediated acetylation of the SMC3 subunit and binding to CTCF [60,61]. The cohesin-SA2 binds to chromatin more slowly. A recent study has shown that this occurs within 2 hours after the completion of division. This isoform is not required for the formation of the first loop structures after the completion of mitosis. Instead, it binds to the cohesin-SA1 loops, already limited by CTCF, and forms smaller nested loops, which is the mechanism of nested loops that was mentioned earlier [58]. These loops have a much shorter life span, since cohesin-SA2 has high sensitivity to the WAPL unloader [61]. It is also likely that these loops determine tissue-specific transcription, as they are often enriched with enhancers and super-enhancers [57].

In addition to forming the structure of interphase chromatin, cohesin also plays an important role in the cohesion and proper segregation of sister chromatids during mitosis. Therefore, cohesin actively binds to chromatin in the S-phase after DNA replication. The molecular mechanisms establishing the cohesion of sister chromatids are not fully clear, but it is believed that this can occur in two independent ways. The “de novo” pathway uses cohesin complexes loaded onto DNA during the S-phase, while the “conversion” pathway uses cohesin previously loaded onto chromatin [62]. But it remains unclear how the replication fork passes through the already loaded cohesin, since due to its size, it is unlikely that it will be able to pass through the cohesin ring. In a recent study, Cameron et al. showed that cohesin is pushed out by the replication fork along the DNA to positions of fork convergence and remains there after the forks are disassembled, capable of binding sister chromatids [63]. In addition, cohesin binding to the protein sororin is necessary to establish cohesion [64], as well as its acetylation by ESCO1 and ESCO2, and according to the recent data, ESCO2 plays a more important role in this process [60]. Together, sororin binding and acetylation protect cohesin from release by the WAPL unloader and prevent premature segregation of sister chromatids.

In mitotic chromosomes, gene expression is mostly suppressed. Therefore, as cells approach division, transcriptional activity decreases and the need for cohesin as a loop extruder and expression regulator disappears. During the transition to the prophase, the structures of compartments and TADs are lost [65]. Complete cohesion of the chromatids is maintained until the prophase. During the prophase, however, mitotic kinases phosphorylate cohesin and sororin, and after this, most of the cohesin is removed from the arms of the chromosomes by the WAPL unloader. Only a small amount of cohesin remains in the centromeric region, where it is protected by the complex of SGO1 and protein phosphatase 2A (PP2A). This complex participates in the process of cohesin and sororin dephosphorylation, which prevents the binding of WAPL to cohesin [66]. Finally, during the anaphase, the RAD21 subunit is cleaved by separase, and the remaining cohesin is removed from the centromeric regions of the chromosomes. This disrupts the cohesion between the sister chromatids, causing them to separate and move to opposite poles of the cell [66,67].

While cohesin is removed from mitotic chromosomes, another component of the loop extrusion machinery, condensins, begins to work. In most metazoans, two condensin isoforms, I and II, are responsible for the condensation of mitotic chromosomes [15]. The binding of these two isoforms to chromatin is separated in time and space [68].

Condensin II is a nuclear complex that binds to chromatin during the prophase, even before the destruction of the nuclear envelope. During the interphase, this complex is also present in the nucleus, but in an inactive form, and is most likely unrelated to chromatin. It has been suggested that condensin II may be involved in loop extrusion and gene regulation during the interphase, like cohesin. But a recent study has disproved this notion. Experiments on the inactivation of condensin II in interphase have not revealed any significant changes to the organization of interphase chromosomes or gene expression. The previous indirect evidence was likely a consequence of mitotic defects affecting the aberrant organization of chromosomes after exiting from mitosis [15]. Condensin II inactivation in the interphase occurs through an unknown mechanism involving MCPHl, preventing its stable binding to DNA. Later, in the prophase, this protein is likely inactivated by phosphorylation, presumably via Cdk1, causing condensin to be able to bind again to chromatin and form mitotic chromosomes [50]. Another study suggested that condensin II may have a self-suppression mechanism which is also inactivated by mitotic phosphorylation from Cdk [69].

The condensin I complex is cytoplasmic and gains access to chromatin only at the end of the prophase, after the destruction of the nuclear envelope. However, recent studies have reported that a small pool of condensin I remains bound to chromatin after the completion of division and thus enters the nucleus, where it remains during the interphase [58]. However, it is unlikely that condensin I is involved in the loop extrusion during interphase, since its function and loading onto chromatin require binding to KIF4A and mitotic phosphorylation [49,70]. But there are exceptions. It has been shown that *C. elegans* condensin I, together with cohesin, are the main loop extruders of interphase chromatin. Their loss leads to the decompression of the entire chromatin, disappearance of TADs, and mixing of chromosomes [71].

The formation of mitotic chromosomes by two condensins occurs according to the nested loop mechanism mentioned previously [65]. Condensin II binds to chromatin first and forms long loops along the axis of the chromosome, shortening it longitudinally. The absence of condensin II in the cell leads to the formation of elongated zigzag-like chromosomes without a defined axis. After the nuclear envelope breaks down, condensin I binds to chromatin and forms shorter nested loops on the long loops elongated by condensin II, due to which the width of the chromosome decreases. Chromosomes without condensin I have fuzzy and fluffed outlines [72,73]. Condensation of chromatin by condensins gives mitotic chromosomes rigidity, especially at the centromere, which is necessary to resist the pulling forces from the division spindle and proper segregation [74].

After the chromosomes diverge during the anaphase, condensins lose their ability to bind to chromatin, and most of them are removed from the chromosomes, waiting for the next cell division cycle. An intermediate state with a minimum number of SMC complexes is formed during the telophase [54].

Despite the fact that condensin II does not directly participate in the organization of chromosomes in the interphase, it has been shown to play a key role in regulating chromatin architecture and the formation of chromosomal territories. During mitosis, condensin II extrudes loops at centromeres, which likely leads to a reduction in the surface area of heterochromatin and limits the interaction between centromeres. This prevents the clustering of centromeres and subsequently leads to the formation of chromosome territories in the interphase nucleus [75].

Another member of the SMC family of complexes, SMC5/6, has been studied less than cohesin and condensins, and its functions are not yet fully understood [76]. Nevertheless, based on recent data obtained, it appears that it is also necessary for normal cell function. Disruption of SMC5/6 functions leads to the induction of DNA damage, p53 activation, cell-cycle arrest, and various mitotic defects such as lagging chromosomes and large DNA bridges. A recent study [77] found that SMC5/6 is associated with chromatin during the G1, S, and G2 phases, as well as during mitosis. Additionally, it has been shown that SMC5/6 plays an important role in the proper segregation of chromosomes during the interphase. The inactivation of SMC 5/6 in the S-phase leads to serious mitotic aberrations, while its inactivation at the end of the G2-phase or mitosis does not affect the phenotype during mitosis [77]. Presumably, SMC5/6 is involved in the resolution of repair intermediates, which in turn ensures the entry of repaired and replicated DNA into mitosis [78,79,80]. But why the complex remains bound to chromatin during mitosis is not fully understood and requires further study.

Thus, the loading and removal of loop extrusion machinery components throughout the cell cycle play a key role in regulating gene expression, in forming chromatin architecture, and in the formation and segregation of mitotic chromosomes.

## 4. Mutations of the Loop Extrusion Machinery

When analyzing the literature, it seems that among the SMC complexes, the greatest attention is paid to cohesin, then condensins, and the least data exist on SMC5/6. The same ratio is characteristic of our review. It is possible that such a distribution reflects not just a tradition, but the “importance” of the functions performed by SMC complexes. As a metric of such importance, one can use the LOEUF (loss-of-function observed/expected upper bound fraction) score, which shows the loss-of-function constraint for individual genes. It is interesting that the genes of proteins involved in the work of SMC complexes have different LOEUF scores (Figure 3). Thus, all the key genes involved in the work of cohesin have LOEUF scores less than 0.6—a threshold for genes that are essential for human cell viability [81]. All genes of the SMC5/6 complex have LOEUF scores greater than 0.6. And the condensin genes occupy an intermediate position. Thus, the LOEUF score indirectly confirms the different “importance” of SMC complexes. Note, however, that this is based on predicted loss-of-function variants, whereas pathogenic variants may not only be loss-of-function.

## 5. Germline Mutations

Loop extrusion machinery participates in a range of processes essential for cell viability, such as spatial chromatin architecture, transcriptional regulation, DNA repair, and sister chromatid cohesion. Hence, germline pathogenic variants of genes encoding subunits of the SMC complexes and their interacting proteins are known to cause developmental disorders characterized by intellectual disability, growth retardation, craniofacial dysmorphism, limb defects, or gonad failure in the case of meiotic subunits. Table 1 shows a list of germline disease-associated mutations of the SMC complexes.

It may be expected for the mutations in SMC complexes to cause sister chromatid cohesion failure, but studies of Cornelia de Lange syndrome (CdLS) have revealed that this is not necessarily the case. CdLS is characterized by a certain degree of clinical variability. Its manifestations include distinctive facial features, growth impairment, limb defects, multi-organ pathology, and frequent intellectual deficits [82]. CdLS is perhaps the most well-studied disease associated with pathogenic variants of the cohesin subunits, and its five proteins (NIPBL, SMC1A, SMC3, HDAC8, and RAD21) are the major contributors to CdLS [83,84]. Mutations in *NIPBL* are the primary genetic cause of CdLS (>60% cases), being associated with the severe and classic forms of CdLS [85]. In addition to a range of *NIPBL* variants, a variant of its binding partner, *MAU2*, has been described to cause a CdLS phenotype [86]. The *MAU2* mutation interrupts a site for interaction between NIPBL and MAU2; thus, a functional cohesin loading complex is unable to form. Similarly, N-terminally truncated NIPBL also leads to the inability to form a complex. However, surprisingly, both impairments do not affect cohesion or cohesin binding. NIPBL is able to mediate cohesin loading onto chromatin, although a lower cohesin occupancy was observed, consistent with the reduction in the chromatin-bound NIPBL fraction. The authors propose that it could be adopted as a protective mechanism, that is, alternative translation initiation could yield a form of NIPBL missing N-terminal residues, which enables cohesin binding.

Although cohesin binding and cohesion are normal, gene expression is impaired in CdLS patients. The transcriptional profiles of cortical neurons from CdLS patients revealed deregulation of genes with neuronal functions related to synaptic transmission, signaling processes, learning, and behavior [87]. In agreement with this, transcriptome analysis of *NIPBL*-mutated CdLS patient-derived primary fibroblasts revealed the downregulation of genes involved in development and system skeletal organization, which was also detected for the *SMC1A*-mutated CdLS [88]. In addition, cohesin was observed to accumulate at NIPBL-occupied sites where cohesin is likely loaded, rather than at the cohesin-CTCF sites where cohesin’s final destination is. Since it has been suggested that NIPBL participates in not only cohesin loading, but also cohesin translocation alongside chromatin [89], NIPBL’s post-loading function is believed to explain extensive gene expression changes and global misplacement of cohesin. Another non-cohesion-related function in gene regulation has been proposed for NIPBL due to its binding to transcriptional regulator BRD4 [90], impairments of which, among other transcriptional factors, could compromise NIPBL binding to H3K27ac-associated active chromatin and, consequently, cohesin loading in early G1 [59].

Thus, *NIPBL* mutations in CdLS patients are associated with gene misregulation rather than cohesin binding or cohesion failure. This might be critical in early development, which requires a precise gene expression tuned for appropriate timing in a process of rapidly changing different regulator patterns. This is similar to the fact that misregulation of epigenetic landscape during neural crest development could compromise precise gene expression and, consequently, contribute to congenital birth defects affecting the craniofacial skeleton [91].

Although we discussed only one example of SMC complex dysfunction as a result of germline mutations, it unveils that these complexes can contribute to pathogenesis independent of chromosome shaping and cohesion.

## 6. Somatic Mutations

Sequencing of cancer genomes revealed a wide array of somatic mutations affecting SMC core subunits and their regulators. The impacts of different SMC family mutations are complex and sometimes contradictory given their versatile roles in sister chromatid cohesion, transcriptional regulation, and DNA damage repair.

### 6.1. Cohesin

Cohesin is the most well-studied SMC protein in terms of its association with cancer [92]. Somatic mutations in genes encoding cohesin are common in bladder cancer (10% frequency in muscle-invasive and up to 40% in non-muscle-invasive variant) [93,94], colorectal cancer (20%) [95,96,97], Ewing sarcoma (15–20%) [98], myeloid malignancies (20%) [99,100], endometrial cancer (19%) [101], glioblastoma (7%) [102], and other tumor types [103]. As indicated earlier, two somatic cohesin variants coexist in cells—cohesin-SA1 and SA2 [104]. They contribute differently to loop formation and gene expression [57] and have distinct roles in sister chromatid cohesion, where cohesin-SA1 is required for cohesion along telomeres and cohesin-SA2 is crucial for centromere cohesion [105,106]. Interestingly, two cohesins differentially contribute to double-strand break (DSB) repair. Cohesin-SA2, but not SA1, has been shown to preferentially associate with DSB sites and facilitate sister chromatid homologous recombination repair over NHEJ [107].

Previous studies have proposed that downregulation of cohesin subunits or its regulators leads to aberrant chromosome segregation in cell cultures [96,108,109] and in model organisms [9,110,111].

### 6.2. Cohesin Mutations and Aneuploidy

Frequent mutations in the cohesin genes found in different tumors urged us to resolve the molecular mechanism behind these observations. Early investigations in the field of cohesin cancer biology were focused mostly on chromosome segregation defects and aneuploidy. Analysis of human cancer cell lines revealed frequent inactivating mutations of the *STAG2* gene in glioblastoma, Ewing sarcoma, and melanoma primary tumors. Since *STAG2* is an X-linked gene, its inactivating mutations lead to a loss of function. Correction of the mutant *STAG2* allele in two aneuploid glioblastoma cell lines restored normal chromatid cohesion [112]. Additionally, mutations in another cohesin subunit *SMC1A* were linked to chromosome instability (CIN) in early colorectal adenomas [95]. The phenomenon of CIN is defined as an increased frequency of numerical or structural changes in chromosomal segments or entire chromosomes [113]. In a study by Cucco and colleagues, the frequency of *SMC1A* mutations decreased from early adenomas to colorectal cancers, and the authors suggested that cohesin mutations could stimulate cancer progression by producing a pool of cells with abnormal chromosome content. However, further testing of individual tumor-derived cohesin mutations led to questioning regarding the hypothesis of its role in aneuploidy. Two studies did not reveal associations between mutations in cohesin and aneuploidy in myeloid diseases for the *STAG2*, *RAD21*, *SMC1A,* and *SMC3* subunits [100] and for *RAD21* and *STAG2* [99]. Similarly, a number of works on Ewing sarcoma have also reported no difference in tumor ploidy among tumors expressing *STAG2* or those with *STAG2* loss [98,114,115]. Finally, a subset of tumor-derived *STAG2* mutations was introduced into human cells to analyze the chromosome counts [116]. In this experiment, only one out of seven nonsense mutations in *STAG2* resulted in alterations in chromosome counts, despite the observed aberrations in sister chromatid cohesion. Cells with missense mutations maintained normal sister chromatid cohesion and chromosome counts.

Together, these studies show that cohesin mutations are likely involved in tumorigenesis by mechanisms not limited to or unrelated to chromosome segregation.

### 6.3. STAG2 Mutations Induce Chromatin Rewiring in Cancer Cells

Further discoveries suggest the role of cancer-causing cohesin mutations in 3D genome structure and gene expression. Exhaustive studies using Hi-C have shown that ablation of RAD21 or NIPBL dramatically reduces chromatin looping genome-wide [14,117]. However, testing the effects of *STAG2* inactivation, which is more frequently mutated in cancer, showed more subtle changes in TADs structure. Studies using siRNA or auxin inducible degron systems revealed that STAG2 depletion does not eliminate TADs globally. Instead, according to different experiments, it can reduce the number of TADs by ~30% and decrease their boundary strength [57,118]. Interestingly, *STAG1* inactivation led to even more pronounced contact reshuffling [118]. The plausible reason for the mild effect is that the majority of SA1 and SA2 positions in the genome overlap, with 60-70% of shared sites [61,119]. Thus, although SA1- and SA2-containing complexes do not colocalize at individual binding sites, they could compensate for each other in case of depletion.

However, a subset of SA2-only cohesin positions might be of special interest, as they frequently overlap with active enhancers and promoters [57,118]. Two complementary studies on Ewing sarcoma cell lines, where wild-type *STAG2* was initially genetically ablated, revealed the disruption of a large number of enhancer–promoter interactions in *STAG2* KO cells [120,121]. Moreover, motif enrichment analysis identified a substantial number of altered chromatin contacts that are anchored by EWS/FLI1—an oncogenic fusion transcription factor driving an oncogenic program [122]. This chimeric protein is characteristic of 90% of Ewing sarcoma cases and results from chromosomal translocation [123,124]. Loss of *STAG2* repressed a subset of EWS/FLI1-regulated genes, weakening the corresponding oncogenic program. At the same time, in both studies, an increased migratory potential of mutant cells was observed. Notably, Ewing sarcoma cells tend to demonstrate heterogeneity between EWSR1/FLI1 *high* and EWSR1/FLI1 *low* states, and the latter cells have a stronger propensity to metastasize [125,126]. Finally, the activation of genes associated with adhesion and motility was detected in *STAG2* KO, which may be an outcome of rewiring of chromatin loops [120]. Thus, *STAG2* KO’s effects on migration are likely explained by (a) a lower activity of EWSR1/FLI1, which is associated with a propensity to migrate and invade; and (b) hijacking of transcriptional programs, with the activation of migration-associated genes.

Another insight from the study by Adane and colleagues is that a loss of *STAG2* in Ewing sarcoma cell lines disrupts PRC2-mediated regulation in a subset of genomic regions. These regions show decreased levels of H3K27me3 repressive mark and altered transcription. However, in a recent study performed in glioblastoma cell lines, an opposing effect of *STAG2* mutation was reported [127]. The mutation resulted in the activation of Polycomb signaling. These findings stand in line with previous works indicating that cohesin can negatively regulate Polycomb signaling [128,129]. This contradiction regarding Polycomb-related effects might be due to the tumor specific features. Additionally, upregulation of several growth factors and oncogenes was reported in *STAG2* mutant glioblastoma cells, which were downregulated upon *STAG2* correction [127].

The exploration of how *STAG2* mutations affect the transcriptome and epigenome was expanded to acute myeloid leukemia (AML) [130]. Reduced loop strength and an altered transcriptional program were detected in cells with genetically altered *STAG2* [130]. Specifically, an altered domain structure was spotted at the *HOXA* cluster. This gene cluster lies at the junction of two TADs, allowing for its stepwise regulation from early to late wave genes [12,131,132,133]. The formation of a new, larger domain in *STAG2* mutant cells was associated with the upregulation of several early *HOXA* genes that were included in the transcriptionally active region. Aberrant *HOXA* gene expression was previously detected in different subtypes of AML [134,135]. A recent study extended the research to AML samples with *STAG2* mutations and primary CD34+ hematopoietic stem and progenitor cells (HSPCs) [136]. In line with earlier studies, alterations in short-range regulatory loops were reported, which was associated with downregulation of target genes. Fischer and colleagues showed that the mutation has leukemogenic potential, increasing HSC self-renewal and reducing their differentiation. However, loss of *STAG2* alone is insufficient for the development of leukemia, and *STAG2*/*Runx1*, *STAG2*/*IDH1,* and other combinations are frequent [137,138].

To summarize this section, a loss of *STAG2* function leads to a weakening of enhancer–promoter interactions and altered expression of genes, including those involved in malignant transformation. In addition to the aforementioned studies, STAG2 LOF was shown to affect short-range genomic contacts and related gene expression in bladder cancer cells [139] and melanoma [140]. To date, the largest body of work has focused on the STAG2 subunit. However, two recent studies show that the *SMC1A* variant observed in patients with AML also induces aberrant DNA looping and misexpression of lineage-specifying factors [141,142], emphasizing the need for further investigation.

### 6.4. Cohesin Mutations Affect Alternative Splicing

Dysregulated splicing characterizes many cancer types [143,144]. It was recently shown that intragenic CTCF-mediated chromatin loops can affect the inclusion of the exons [145,146]. And in addition to cohesin’s role in genome organization, it has a direct role in alternative splicing, as evidenced by auxin-inducible depletion of RAD21 [147]. Singh and colleagues have demonstrated that cohesin interacts with core components of the splicing machinery, in line with previous studies [148,149]. Moreover, there was a correlation between cohesin mutations and changes in patterns of alternative splicing in AML patients. Samples with mutated cohesin, but without mutations in any splicing factors, displayed a distinct pattern of alternative splicing that was not observed in AML samples with splicing factor mutations or without mutations in either cohesin or splicing factors. Finally, three out of four tested AML-associated point mutations in *SMC1A* and *SMC3* subunits led to the loss of binding with a splicing factor U1-70. This novel function of cohesin expands our understanding of the progression of cancer.

### 6.5. The Role of Cohesin in DNA Replication

Prior studies have demonstrated that the recruitment of cohesin to the chromatin to concatenate sister chromatids during the S phase is dependent on multiple components of the DNA pre-replication complex, as they physically interact with cohesin [150,151,152]. On the other hand, cohesin affects replication fork processivity [153], specifically cohesin’s acetylation and ubiquitylation [154,155], and deleting cohesin slows replication dynamics [156]. This information was applied in order to understand the consequences of cohesin mutations in cancer. Mondal and colleagues have investigated the effect of *STAG2* depletion and found that it leads to disruption of the interaction between cohesin and the replication machinery and replication fork stalling [157]. *STAG2* KO also induced failure to establish SMC3 acetylation—a post-transcriptional modification required for proper chromatid cohesion. Fork stalling resulted in replication fork collapse, with accumulation of DNA breaks and activation of DNA damage checkpoint signaling. Work by Wu and colleagues further elucidated cohesin’s role in DNA replication and genome integrity by depleting RAD21 [158]. Depletion induced an increase in genome-wide DNA breaks and translocations. A subset of translocation hotspot genes was identified to be co-mutated with cohesin in different cancers. Mapping of Okazaki fragments revealed that translocations are correlated with DNA replication. Expanding upon previous observations, Wu and colleagues showed that normally, cohesin represses early replication origin firing within chromatin loops. Cohesin depletion induces aberrant DNA replication initiation. Extra-early replication results in replication stress and DNA damage, in turn triggering genome instability.

Here, we covered only a part of the research findings collected to date about the role of cohesin as a tumor suppressor. Recently, cohesin was shown to prevent large-scale genome rearrangements by repressing transcription near DNA breaks [159]. Repression promotes accurate repair at DSBs in the vicinity of actively transcribed genes. This feature of cohesin is lost in *SA2* mutant cancer cells. Intersections between cohesin and DSBs repair continue to accumulate, providing an exciting future research direction [160,161,162].

Additionally, several studies have shown overexpression of *RAD21* and *SMC1A* in colorectal carcinoma [97,163,164]. The overexpression of these subunits was associated with poor prognosis. Silencing of *SMC1A* in xenotransplant experiments in mice reduced tumor size and increased survival, highlighting a potential therapeutic target [165].

### 6.6. PDS5 Mutations in Cancer

PDS5 is another important player in regulating chromatin architecture which is essential for cohesin binding to chromatin. Mutated *PDS5* has been found in gastric, colorectal [166], and breast [167,168] cancer. Studies on the mechanisms of the tumor-suppressive role of PDS5 have mostly been focused on its cohesin-independent functions. *PDS5B* upregulation repressed viability, migration, and invasion in studies on lung [169] and pancreatic cancer cells [170]. A recently proposed mechanism of how *PDS5B* depletion might affect the proliferation of cancer cells is through the IL-6/STAT3/cyclin D axis [171]. Downregulation of *PDS5* increases IL-6 secretion and STAT3 activation, which upregulates *cyclin D* expression. It was recently reported that PDS5B positively regulates the expression of *LATS1* in lung cancer cells, a kinase which is known to regulate the cell cycle, cell differentiation, and cell motility [169].

### 6.7. The Role of Mutations in Condensin I and II in Generating CIN

Much less data have been collected to date about the contribution of condensins to cancer development, although mutation frequency analysis across different cancers has revealed comparable numbers with cohesin [172]. It is currently under debate whether such mutations promote cancer progression, but most evidence links condensin dysfunction with CIN. Mutations in *SMC2* and *SMC4* subunits are associated with abnormal chromosomal structures in lymphoma samples, i.e., chromosome bridges and abnormalities in chromosome length and width [173]. Recently, using patient-derived datasets, Baergen and colleagues assessed gene copy number alterations in each condensin subunit gene and identified that deletions occur frequently in 12 common cancer types [174]. An siRNA-based screen was performed to determine the impact of reduced expression of each of eight condensin genes in two karyotypically stable cell lines. Gene silencing was associated with changes in chromosome numbers and CIN-associated phenotype, i.e., micronucleus formation. In a more recent study, three cancer-associated missense mutations in *CAPH2* subunit were investigated [175]. These mutations are located in the SMC4 binding region; thus, they caused abrogated binding to other condensin II subunits. These mutations caused anaphase defects, such as anaphase bridges and lagging chromosomes. Both *CAP-H2* knockdown and overexpression of mutant *CAP-H2* triggered the formation of micronuclei. Finally, the analyzed mutations increased DNA damage, with a significant proportion of the DNA damage foci detected at telomere regions. This is consistent with previous reports stating that depletion of CAP-H2 leads to DNA foci accumulation [176]. In a study by Wallace and colleagues, it was found that NCAPH2 localizes to telomeric repeats and binds to the TRF1 and TRF2 components of a shelterin complex which protects chromosome ends. Depletion of CAP-H2 resulted in the accumulation of telomeric DNA damage foci [176].

Overexpression of condensin subunits has been detected in various tumors, and their knockdown has been shown to inhibit cancer cell proliferation [177,178,179,180,181,182]. However, the mechanism of how this overexpression is involved in tumorigenesis is still unclear.

Together, these findings assume that condensin dysfunction contributes to cancer progression through (a) impaired compaction of chromosomes during cell division and (b) DNA damage at telomeric repeat regions. Eventually, this results in genomic instability, which might provide cells with a selective advantage during cancer progression.

### 6.8. MCPH1 Mutations in Cancer

MCPH1 is a multifunctional protein implicated in the maintenance of genome integrity. It regulates chromosome conformation during the interphase [50] and is involved in cell cycle progression [183], DNA damage response [184], and telomere replication and repair [185]. *MCPH1* is downregulated in different types of human cancers, including colorectal (10%) [186], ovarian (8%) [187], breast (7%) [188], lung (7%) [189], and other cancer types [190]. The molecular effects of MCPH1 dysfunction in cancer are not fully understood. Some studies link its deficiency with defective DNA repair in tumors [191,192]. Mutated *MCPH1* has been shown to induce global transcriptional changes associated with invasion and metastasis in breast epithelial cells [193]. Deletion of *MCPH1* is also associated with centrosome amplification—a common feature in different cancers which correlates with worse outcomes [194].

### 6.9. SMC5/6 Mutations in Cancer

The scope of SMC5/6 complex action includes chromosome replication [195], DNA repair [196,197], and epigenetic silencing of viral DNA [198]. Somatic *SMC5/6* mutations have been seen in many cancers, e.g., ovarian (20%), breast (18%) [199], endometrial (14%), prostate (11%) [200], and other cancer types [201,202]. The causative links between these mutations and disease progression are still under debate, although promising results were obtained in a recent analysis of 65,000 cancer samples across multiple tissues [200]. Alterations in the SMC5/6 complex were associated with ploidy defects and worse survival outcome, while no co-mutational pattern was observed with genes related to genome instability.

In addition, multiple viruses targeted by SMC5/6 activity have tumorigenic potential, e.g., Hepatitis B virus [203], Kaposi’s sarcoma-associated herpesvirus [204], and Epstein–Barr virus [205]. It was recently discovered that Hepatitis B virus Protein X induces SMC5/6 degradation, which leads to the accumulation of DNA damage [206].

## 7. Conclusions

SMC complexes act together to organize and regulate DNA, and each complex is responsible for a unique set of functions. Their essential role is evident from their evolutionary conservativity, where eukaryotic and prokaryotic SMCs share a common principal structure [207,208]. While prokaryotes have one or two SMC complexes, eukaryotes typically possess four, obtained as a result of duplication. This produces subfunctionalization and increases the complexity of their functions. In addition to the core subunits, eukaryotic SMCs have multiple regulators that control their loading/unloading from DNA. Germline and somatic defects in SMC-related genes can lead to various diseases, and ongoing research is underway to define the mechanisms and potential treatments.

## Figures and Tables

**Figure 1 cells-13-01896-f001:**
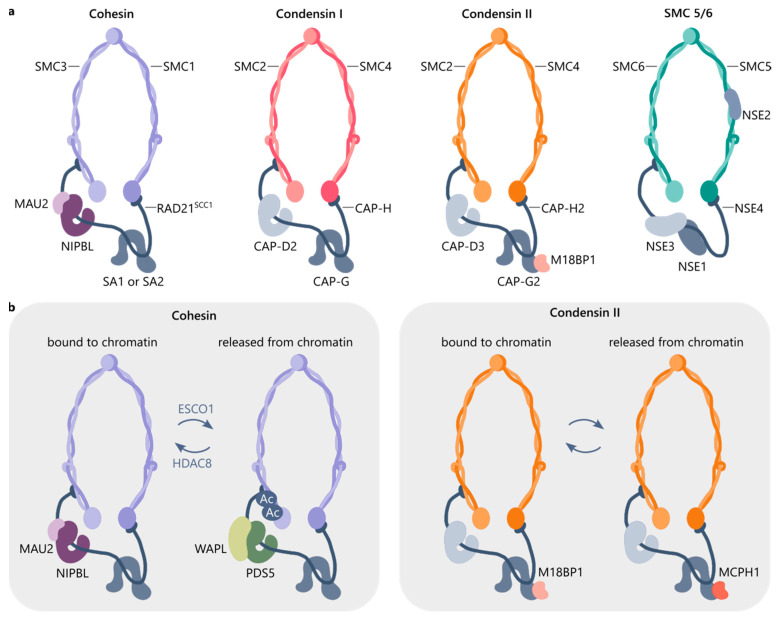
Composition of the loop extrusion protein complexes. (**a**) A dimer of SMC proteins and a kleisin protein (RAD21, CAP-H, CAP-H2, or NSE4) form a tripartite ring. HEAT (SA1 or SA2, CAP-D2, CAP-D3, CAP-G, CAP-G2) or kite (NSE1, NSE3) subunits bind to a kleisin. (**b**) Additional proteins regulate whether a complex is bound to chromatin. An interplay between NIPBL/MAU2 and PDS5/WAPL provides a dynamic balance of extruding cohesin, with ESCO1 facilitating PDS5 binding to cohesin via SMC3 acetylation and HDAC8 limiting it. Similarly, M18BP1 protein facilitates condensin II binding to chromatin, while MCPH1 restricts it.

**Figure 2 cells-13-01896-f002:**
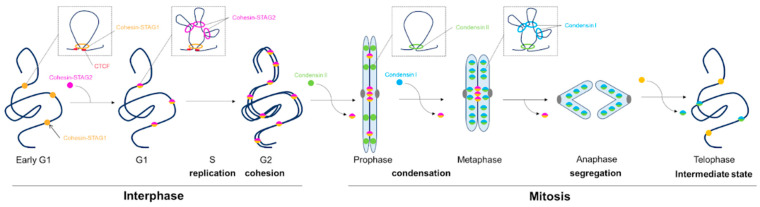
Dynamics of cohesin and condensins in cell cycle.

**Figure 3 cells-13-01896-f003:**

Distribution of LOEUF score for genes involved in SMC complexes. Blue violin plot shows the distribution of LOEUF scores for all human genes from the gnomAD v4.1 database. The red line marks the threshold of LOEUF score < 0.6 for genes that are essential for human cell viability.

**Table 1 cells-13-01896-t001:** List of diseases associated with germline mutations of the SMC complexes and their regulatory proteins.

Gene	Function	Associated diseases
Cohesin		
*SMC1A*	Cohesin core subunit	CdLS (OMIM #300590), developmental and epileptic encephalopathy (OMIM #301044)
*SMC3*	Cohesin core subunit	CdLS (OMIM #610759)
*RAD21*	Cohesin core subunit	CdLS (OMIM #614701)
*STAG1*	Cohesin core subunit	Intellectual developmental disorder (OMIM #617635)
*STAG2*	Cohesin core subunit	Holoprosencephaly (OMIM #301043), Mullegama–Klein–Martinez syndrome (OMIM #301022)
*STAG3*	Meiotic cohesin core subunit	Premature ovarian failure (OMIM #615723), spermatogenic failure (OMIM #619672)
*NIPBL*	Cohesin loading	CdLS (OMIM #122470)
*HDAC8*	Deacetylation of Smc3	CdLS (OMIM #300882)
Condensins		
*NCAPG2*	Condensin II core subunit	Khan–Khan–Katsanis syndrome (OMIM #618460)
*NCAPD3*	Condensin II core subunit	Microcephaly (OMIM #617984)
*MCPH1*	Condensin II dissociation	Microcephaly (OMIM #251200)
SMC5/6		
*SMC5*	SMC5/6 core subunit	Atelis syndrome (OMIM #620185)
*NSE2*	SMC5/6 core subunit	Seckel syndrome (OMIM #617253)
*NSE3*	SMC5/6 core subunit	Lung disease, immunodeficiency, and chromosome breakage syndrome (OMIM #617241)

## Data Availability

Not applicable.

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
