# Peer review of "Loop Extrusion Machinery Impairments in Models and Disease"

_cells, 2024, doi:10.3390/cells13221896_

Round 1
Reviewer 1 Report
Comments and Suggestions for Authors
This is an excellent short review of SMC complexes and their effect when mutated in model organisms or in disease. The numerous references are a very useful collection for further study, and the description of current knowledge of each of these complexes is well worth reading. While the disease descriptions are short, there are enough references to investigate them individually in further detail. The English is clear and I see no corrections necessary. I've found this manuscript interesting to read, and I have no doubt it will be useful reading for a Cells audience.
Author Response
We thank the Reviewer for an appreciation of our review paper. There were no comments to address.
Reviewer 2 Report
Comments and Suggestions for Authors
The review by Ryzhkova et al. reviews SMC complexes cohesin, condensins and SMC5/6 in loop extrusion and in disease. It starts with experimental methods to manipulate these complexes, then the structure and function of these complexes. It then describes the dynamics of these complexes in cell cycle and how the function of loop extrusion is passed from one complex to another from interphase to prophase and the different stages of mitosis. Following the normal function of these complexes, the authors reviewed germline and somatic mutations and their consequences on diseases.
Overall, the review is well written and quite comprehensive. Although there are many reviews on SMC complexes, the section on cell cycle is one of the clearest sections written about SMC complexes in cell cycle. I only have minor comments.
Lines 97-100. PROTACs being used for cancer therapy is not relevant for this review and can be removed.
Fig 3. Condensin is spelled wrongly
Section 5. The phenotype of CdLS should be briefly described since the whole section is on CdLS.
Line 557. Define CIN
Section 6.8 I think this should be grouped near the condensin section (6.6)
Section 6.9 I think this should be grouped near the cohesin section (6.1-6.5)
References 101 and 102 are the same
Author Response
Comment 1: Lines 97-100. PROTACs being used for cancer therapy is not relevant for this review and can be removed.
Response 1: Thank you for the comment. According to your suggestion, we have removed information about the PROTACs usage in therapy.
Comment 2: Fig 3. Condensin is spelled wrongly
Response 2: Thank you for pointing this out. We have modified Figure 3 to correct the misspelling.
Comment 3: Section 5. The phenotype of CdLS should be briefly described since the whole section is on CdLS.
Response 3: Thank you! We have outlined some clinical manifestations of the disease in Section 5: "CdLS is characterized by a certain degree of clinical variability. Its manifestations include distinctive facial features, growth impairment, limb defects, multi-organ pathology and frequent intellectual deficits".
Comment 4: Line 557. Define CIN
Response 4: Thank you for drawing our attenion to the problem. We have to point out that we defined CIN earlier in the text, at line 411: “Additionally, mutations in another cohesin subunit SMC1A was linked to the chromosome instability (CIN) in early colorectal adenomas”. But due to your suggestion we also additionally provided a brief description of CIN: "The phenomenon of CIN is defined as an increased frequency of numerical or structural changes in chromosomal segments or entire chromosomes". We hope that this clarifies the definition of CIN.
Comment 5: Section 6.8 I think this should be grouped near the condensin section (6.6)
Response 5: Thank you for the suggestion. We have moved the section “MCPH1 mutations in cancer” (current 6.8) before the section “SMC5/6 mutations in cancer” (current 6.9).
Comment 6: Section 6.9 I think this should be grouped near the cohesin section (6.1-6.5)
Response 6: Thank you for the suggestion. We have moved the section “PDS5 mutations in cancer” (current 6.6) near the cohesin section, before the section “The role of mutations in Condensin I and II in generating CIN” (current 6.7).
Comment 7: References 101 and 102 are the same
Response 7: Thank you for the comment. We have revised the Reference list to ensure there is no repetition.